# Novel Anti-Melanoma Compounds Are Efficacious in A375 Cell Line Xenograft Melanoma Model in Nude Mice

**DOI:** 10.3390/biom13091276

**Published:** 2023-08-22

**Authors:** Sadeeshkumar Velayutham, Ryan Seerattan, Maab Sultan, Trisha Seal, Samaya Danthurthy, Baskaran Chinnappan, Jessica Landi, Kaitlyn Pearl, Aveta Singh, Keiran S. M. Smalley, Julia Zaias, Jun Yong Choi, Dmitriy Minond

**Affiliations:** 1College of Pharmacy, Nova Southeastern University, 3321 College Avenue, Fort Lauderdale, FL 33314, USA; 2Rumbaugh-Goodwin Institute for Cancer Research, Nova Southeastern University, 3321 College Avenue, CCR r.605, Fort Lauderdale, FL 33314, USA; ms4323@mynsu.nova.edu; 3Department of Chemistry and Biochemistry, Queens College, 65-30 Kissena Boulevard, Flushing, NY 11367, USA; 4Halmos College of Arts and Sciences, Nova Southeastern University, 3301 College Avenue, Fort Lauderdale, FL 33314, USA; 5Honors College, Nova Southeastern University, 8000 N Ocean Dr., Dania Beach, FL 33004, USA; 6Dr. Kiran C. Patel College of Osteopathic Medicine, Nova Southeastern University, 3321 College Avenue, Fort Lauderdale, FL 33314, USA; 7Department of Tumor Biology, Moffitt Cancer Center, 12902 Magnolia Drive, Tampa, FL 33612, USA; keiran.smalley@moffitt.org; 8Division of Comparative Pathology, University of Miami, 1501 NW 10th Ave, Miami, FL 33136, USA; jzaias@med.miami.edu; 9Ph.D. Programs in Chemistry and Biochemistry, The Graduate Center of the City University of New York, 365 5th Ave, New York, NY 10016, USA

**Keywords:** melanoma, drug discovery, spliceosomal inhibition, BRAF, cell line xenograft, organ histopathology

## Abstract

Despite the successes of immunotherapy, melanoma remains one of the deadliest cancers, therefore, the need for innovation remains high. We previously reported anti-melanoma compounds that work by downregulating spliceosomal proteins hnRNPH1 and H2. In a separate study, we reported that these compounds were non-toxic to Balb/C mice at 50 mg/kg suggesting their utility in in vivo studies. In the present study, we aimed to assess the efficacy of these compounds by testing them in A375 cell-line xenograft in nude athymic mice. Animals were randomized into four groups (n = 12/group): 10 mg/kg vemurafenib, and 25 mg/kg 2155-14 and 2155-18 thrice a week for 15 days along with a control group. The results revealed that both 2155-14 and 2155-18 significantly decreased the growth of A375 tumors, which was comparable to vemurafenib. These results were confirmed by tumor volume, weight, and histopathological examination. In conclusion, these results demonstrate the therapeutic potential of targeting spliceosomal proteins hnRNPH1 and H2.

## 1. Introduction

Melanoma is the deadliest form of skin cancer responsible for more than 7000 deaths a year in the US alone [1]. Before approval of immunotherapy, melanoma patients with metastatic disease had a 5-year survival rate of approximately 20% [2]. Immunotherapy alone or in combination with small molecule drugs improved overall 5-year survival (OS) to 50% with approximately 40% of patients responding to the therapy [3,4]. However, adverse effects of immunotherapy, especially the combination of anti-CTL-4 and anti-PD-1 treatment, often lead to the stoppage of the treatment [5,6]. Additionally, a portion of the patients who initially respond to the therapy eventually acquire resistance [7]. Based on these considerations novel approaches to melanoma therapy are needed.

We previously reported the discovery of spliceosome-binding small molecules active against melanoma in vitro [8,9] and non-toxic to mice [10]. These molecules, namely 2155-14 and 2155-18, were discovered via a mixture-based phenotypic high throughput screen (HTS) using multiple cancer and non-cancer cell lines [9]. Both compounds demonstrated low micromolar EC_50_ values for inhibition of the viability of multiple melanoma cell lines while sparing the viability of several cell lines representative of various cancers and non-cancer cell lines. We demonstrated that the mechanism of action of these molecules is based on the binding of hnRNP H1 and H2 which are RNA-binding proteins (RBPs) with a role in pre-mRNA splicing [11]. RBPs have attracted attention as targets for cancer drug discovery due to being differentially expressed in many cancers [12]. There have been multiple reports of RBP inhibitors (RBPIs) targeting various RBPs [12]. Several small molecules targeting eIF4A have been reported by others. Most pertinent to the present study, DMDA-Pat A, a structural analogue of the marine natural product pateamine A, demonstrated in vivo activity against MDA-MB-435 cell-derived melanoma xenograft (CDX) at 0.9 mg/kg bw in mice [13]. Small molecule 4EGI-1, an inhibitor of RBP eIF4E/G with Ki of 25 µM, inhibited growth of CRL-2813 melanoma mouse CDX at 25 and 75 mg/kg q.d. [14]. Remarkably, neither inhibitor exhibited overt toxicity at the tested doses.

Both eIF4A and eIF4E/G are involved in mRNA translation, however, there are examples of small molecules targeting other RNA-related processes. Small molecules E7107, Spliceostatin A, and H3B-8800 target mRNA splicing by modulating Splicing Factor 3B1 (SF3B1). E7107 demonstrated efficacy in a pre-clinical model of NSCLC at 20 mg/kg; however, it exhibited dose-dependent toxicity in two Phase I clinical trials [15,16] (MTD = 4 mg/m^2^) which led to the discontinuation of this trial. H3B-8800 treatment was associated with mostly low-grade adverse effects after 1–40 mg once-daily dose [17] in Phase I trial in patients with myelodysplastic syndromes (MDS), chronic myelomonocytic leukemia (CMML), and secondary acute myeloid leukemia (AML) arising from MDS.

In the present study, we tested the efficacy of two small molecules that bind and downregulate spliceosomal proteins hnRNP H1 and H2 in a cell-derived xenograft mouse model of BRAF-mutant melanoma.

## 2. Materials and Methods

Procedure for the synthesis of 2155-14 (JC-395) and 2155-18 (JC-408). Pyrrolidine-bis-diketopiperazine JC-395 and JC-408 (Figure 1A) were synthesized by modifying the previously published method [9]. Both compounds were synthesized via solid-phase methodology (Figure 1) on 4-methylbenzhydrylamine hydrochloride resin (MBHA) (1.4 mmol/g, 100–200 mesh). To a syringe with a solid filter at the bottom was added MBHA resin (150 mg, 1 eq.), which was swelled and neutralized using 10% diisopropylethylamine (DIEA)/dimethylformamide (DMF) (*v*/*v*) (4 mL) at 25 °C for 20 min. After draining the swelling solution, the resin was washed with DCM (2×), Methanol (2×), DMF (2×), and then DCM (2×). [Coupling reaction with amino acid] To an empty glass vial was added Boc-protected amino acid (Boc-Phe-OH, 4 eq.), 1-[Bis(dimethylamino)methylene]-1H-1,2,3-triazolo [4,5-b]pyridinium 3-oxide hexafluorophosphate (HATU, 4 eq.), and hydroxybenzotriazole hydrate (HOBt, 4 eq.), and 10% DIEA/DMF was added to the mixture, which was shaken for 15 min for pre-activation. The pre-activated solution was added to the syringe containing the neutralized resin, and the syringe was shaken for 4 h at 25 °C. The solution was drained from the syringe, and the resin was washed with DCM (2×), Methanol (2×), DMF (2×), and DCM (2×). Around 10–15 beads of resin were taken for the Kaiser test. [Boc deprotection and neutralization] Boc protecting groups were removed with 55% trifluoroacetic acid (TFA)/45% dichloromethane (DCM) (1×, 4 mL, 30 min) and subsequently neutralized with 10% diisopropylethylamine (DIEA)/90% DCM (3×, 5 min). The steps for the coupling reaction with amino acid, Boc-deprotection, and neutralization were repeated with different amino acids such as Boc-Pro-OH, Boc-β-cyclohexyl-D-alanine-OH (or Boc-Phe-OH for JC-408), and Fmoc-Tyr(OtBu)-OH in sequence: the reaction time for coupling with amino acids was 2 h. The terminal Fmoc group was deprotected with 20% piperidine in DMF (4 mL) by shaking at 25 °C for 30 min. After draining the solution and wash the resin with DCM (2×), Methanol (2×), DMF (2×), then DCM (2×), phenylacetic acid (4 eq.) or 2-(Adamantan-1-yl)acetic acid (4 eq.) for compound 18, HATU, (4 eq.), HOBt (4 eq.) in 10%DIEA/90%DMF (4 mL) were pre-activated in a glass vial for 15 min and added to the syringe for shaking at 25 °C for 2 h. After the washing step, the resin was transferred to a microwave vial, and Borane-THF (1M, 6 mL, ~30 eq.) was added to the vial. The reaction mixture was heated under microwave irradiation at 70 °C for 6 h, and the solution was poured off. Piperidine (6 mL) was added to the microwave vial containing the resin, which was heated under microwave irradiation at 70 °C for 30 min. Piperidine was poured off, and the resin was transferred to a clean syringe, which was washed with DCM (2×), Methanol (2×), DMF (2×), and then DCM (2×). Diketopiperazine cyclization was performed under anhydrous conditions. The resin in the syringe was placed in a pressure relief scintillation vial and a solution of 1,1′-oxalyldiimidazole (5× for each cyclization site) in anhydrous DMF (4 mL) was added to the vial, which was stirred at 25 °C for 48 h. After draining the solution, the resin was washed with DCM (2×), Methanol (2×), DMF (2×), then DCM (2×). Completion of cyclization was checked by cleaving a control sample, which was analyzed by LC. The compounds were then cleaved from the resin with trifluoroacetic acid (TFA)/trifluoromethanesulfonic acid (TFMSA) (9:1, 4 mL). The cleavage solution was collected and removed by blowing out N_2_ gas. The crude samples were diluted with MeOH (4 mL) and filtered for HPLC purification as described below to produce the title compound (JC-395 or JC-408) as a white powdery solid with 98% or 99% purity, respectively. 1H NMR and Mass spectroscopic data were matched with those in the previous report [9].

Compound purification and characterization. The final compounds were purified using preparative HPLC with a dual pump Shimadzu LC-20AP system equipped with a SunFire C18 preparative column (19 × 250 mm, 10 microns) at λ = 220 nm, with a mobile phase of (A) H_2_O (0.1% TFA)/(B) methanol (MeOH)/acetonitrile (ACN) (3:1) (0.1% TFA), at a flow rate of 60 mL/min with 10% (B) for 30 sec, a gradient up to 90% (B) for 9.5 min, and 90% (B) for 3 min. ^1^H NMR and ^13^C NMR spectra were recorded in DMSO-*d_6_* on a Bruker Ascend 400 MHz spectrometer at 400.14 and 100.62 MHz, respectively, and mass spectra were recorded using an Advion Mass Express. The purities of the synthesized compounds were confirmed to be greater than 95% by liquid chromatography on a Shimadzu LC-20AD instrument with SPD-20A. The mobile phase of (A) H_2_O (0.1% formic acid)/(B) ACN (0.1% formic acid) was used with a gradient of 5–95% over 7 min followed by 3 min rinse and 3 min equilibration.

Animal protocol. This study used 5- to 7-week-old male and female athymic Nu/Nu mice (The Jackson Laboratory, Bar Harbor, ME, USA). The mice were housed in the standard mouse shoe-box cages and maintained in a 12 h light/12 h dark cycle, with 50% humidity and 20 ± 3 °C. The mice had free access to a standard pellet diet (Certified PicoLab^®^ Rodent Diet 20, Lab Diet) and water ad libitum. The study was conducted in accordance with the guidelines of the Nova Southeastern University (NSU) Institutional Animal Care and Use Committee (NSU IACUC protocol 2019.12.DM4).

The animals were allowed to acclimate after delivery. The animals were randomly divided into 5 groups with each group containing 12 mice (6 male and 6 female). Group 1: non-tumor control mice; Group 2: animals were treated with vehicle control (10%/90% DMSO/sterile water, USP sterile injectable grade); Group 3: animals were treated with vemurafenib (10 mg/kg body weight); Group 4: animals were treated with 2155-14 (25 mg/kg body weight); Group 5: animals were treated with 2155-18 (25 mg/kg body weight) three times/week.

To establish a cell line-derived xenograft (CDX), A375 (ATCC^®^ CRL-1619, ATCC, Manassas, MD, USA) cells were cultured in DMEM media supplemented with 10% FBS, 1% Pen/Strep. On the day of xenograft implantation, the cells were harvested at 70–80% confluency and suspended in 10 mL of sterile PBS (USP grade, sterile for injection) so that 200 µL contained the required number of cells per injection and kept on ice. 2.0 × 10^6^ cells/0.2 mL were injected into the right flanks of appropriate groups of mice using 1 mL insulin syringes with 26-gauge needles.

Mice injected with A375 cells were palpated every day to detect the tumor growth. Once tumors were palpable, the digital caliper was used to measure the width and length of the tumors. Tumor volumes were calculated using the Equation (1):Volume = (width)^2^ × length/2(1)

Compounds (vemurafenib, 2155-14, and 2155-18) were prepared in 10% DMSO/H_2_O (both USP injectable grade) fresh for each treatment day. Vemurafenib, 2155-14, and 2155-18 were weighed into autoclaved 1.5 mL Eppendorf vials using analytical scales. USP grade DMSO was added to each vial under aseptic conditions and vortexed. USP-grade injectable sterile H_2_O was then added to each vial and again vortexed. One milliliter insulin syringes with a 26-gauge needle were filled with 0.2 mL of the compound and delivered to the vivarium in the closed carrier for animal treatment. For the vehicle control group, syringes were filled with 0.2 mL of 10% DMSO/H_2_O (USP injectable grade). All compounds were injected subcutaneously (sc).

During the experimental period, body weights were measured, and mice were observed for signs of clinical distress every day. More specifically, mice were observed for posture, vocalization, ease of handling, lacrimation, chromodacryorrhea, salivation, coat condition, unsupported rearing, arousal, piloerection, motor movements, diarrhea, tail pinch reaction, and constipation.

When tumors reached 2000 mm^3^, the mice were euthanized, tumors excised, weighed, and their volumes were measured again.

The excised tumors were placed in either 10% neutral buffered formalin for paraffin embedding or at −80 °C for cryo sectioning for tumor histopathology. The tissues were processed via standard tissue processing to produce H&E slides. All H&E slides were reviewed blind to the treatment group. The tumors have also been fluorescently stained for cleaved caspase 3 to detect apoptosis, Ki67 for proliferation, and hnRNPH1/H2 to confirm target modulation. The histopathology analyses were performed at the Division of Comparative Pathology, University of Miami.

Statistical significance was set at *p* < 0.05. All data were analyzed using one-way ANOVA to compare means, and significant differences were further analyzed by Tukey’s multiple comparisons using Prism (version 8.0, GraphPad Inc., San Diego, CA, USA).

hnRNP H staining. Tissue sections stored at −80 °C were thawed and fixed for 10 min in 4% paraformaldehyde on ice. Samples were washed in PBS and incubated overnight in 30% sucrose in PBS at 4 °C. Dissected tissue samples were embedded in Tissue-Tek optimal cutting temperature (OCT) compound (Sakura 4583) and stored at −80 °C. Sections of 10 µm thickness were cut using the Leica CM1850 UV Cryostat. Sections were washed in PBS for 10 min at RT, Sections treated with 0.1% sodium borohydride in PBS for 10 min, permeabilized in 0.3% Triton X-100 in PBS for 5 min at RT, and washed in 0.1% Tween20 in TBS. Sections were incubated in blocking buffer (1% BSA, 10% Goat serum, 0.3% triton X100 in PBS) for 1 h at RT. The primary antibody for hnRNP H (rabbit monoclonal, 1:250, abcam ab289974) was incubated overnight at 4 °C in antibody diluent. Sections were washed three times using 0.1% Tween20 in PBS at RT and incubated with secondary antibody (goat anti-rabbit IgG conjugated to Alexa Fluor 647 (ab150079, abcam) diluted 1:400 in antibody diluent for 1 h at RT. Sections were washed three times for 5 min at RT in PBS and mounted in VECTASHIELD Antifade Mounting Medium with DAPI for fluorescence (VECTOR). Images were taken using Evos (Life Technologies, Carlsbad, CA, USA) automated microscope.

Sections were stored short-term at −20 °C and stained with hematoxylin and eosin (H&E, VWR US Cat. No. 95057-844 and VWR US Cat. No. 95057-848) for histopathological examination.

Cleaved caspase 3 staining. Tissue samples were dissected and fixed in 4% paraformaldehyde at RT. Once fixed, the tissues were processed overnight on the Leica ASP 300S Processor (Leica Biosystems, GmbH, Nußloch, Germany) and embedded in paraffin. Slides were sectioned at 5µm and stained using the Leica Bond RXm automated research stainer (Leica Biosystems, GmbH, Nußloch, Germany). Slides were baked (60 °C), deparaffinized (BOND Dewax deparaffinization solution), underwent epitope retrieval (BOND Epitope Retrieval Solution 1), stained with Anti-Cleaved Caspase 3 (Cell Signaling cat# 9664S, 1:200) in combination with the BOND Polymer Refine HRP Plex Detection kit (Leica cat# DS9914), and counterstained with hematoxylin.

Ki67 staining. Tissue samples were dissected and fixed in 4% paraformaldehyde at RT. Once fixed, the tissues were processed overnight on the Leica ASP 300S Processor (Leica Biosystems, GmbH, Nußloch, Germany) and embedded in paraffin. Slides were sectioned at 5µm and stained using the Leica Bond RXm automated research stainer (Leica Biosystems, GmbH, Nußloch, Germany). Slides were baked (60 °C), deparaffinized (BOND Dewax deparaffinization solution), underwent epitope retrieval (BOND Epitope Retrieval Solution 1), stained with Anti-Ki67 (Cell Signaling cat# 12202S, 1:400) in combination with the BOND Polymer Refine HRP Plex Detection kit (Leica cat# DS9914), and counterstained with hematoxylin.

## 3. Results

Synthesis and characterization of 2155-14 and 2155-18. The pyrrolidine diketopiperazine compounds, 2155-14 (JC-395) and 2155-18 (JC-408), were synthesized by standard solid phase synthesis with MBHA resins. Amino acids were coupled to amine on the resin using Boc-AA-OH or Fmoc-AA-OH (4 eq.), HOBt (4 eq.), and HATU (4 eq.) in DMF containing 10% DIEA at 25 °C, and the Boc or Fmoc group was deprotected by 55% TFA in DCM or 20% piperidine in DMF at 25 °C, respectively. After coupling reactions with four amino acids and phenylacetic acid or 2-(adamantan-1-yl)acetic acid, five amide groups were reduced with borane in THF (~30 eq.) at 70 °C under μwave irradiation for 6 h. The reduced 2° amine groups were coupled with 1,1′-oxalyldiimidazole in anhydrous DMF, and diketopiperazine moieties were formed in the agent on the resin. TFA/TFMSA (9/1) was used to cleave the resin, and the final compounds were obtained by preparative HPLC purification with over 98% purity confirmed by the analytical LC. The structures of the final compounds were confirmed by ^1^H NMR and Mass spectroscopic analysis, which were matched with the previous report [9].

Expression of hnRNP H1 and H2 correlates with survival and later disease stages. Since nothing was known about H1/H2 role in patient survival, we explored clinical databases to gain more insight. Analysis of the gene expression data set of 214 metastatic melanoma patients available in the R2 database [18] revealed a correlation between high expression of hnRNP H1 and H2 with low disease-specific survival probability (Figure 1A). Additionally, when stratified by disease stage, hnRNP H1 had greater expression at stages I–IV as compared to stage 0 suggesting a correlation with advanced disease (Figure 1B). This suggests that the decrease of expression of H1 and H2 can be beneficial for patient survival, therefore, treatment with compounds that downregulate the expression of these proteins represents a feasible therapeutic strategy.

In vivo efficacy study in A375 CDX. In the previous study, we reported that 2155-14 and 2155-18 did not cause organ or overt toxicity when tested at 50 mg/kg/day in Balb/C mice [10], which are immunocompetent. In the present study, we utilize immunocompromised mice, therefore, we monitored for overt signs of distress every day. Additionally, as a precaution, we used 25 mg/kg bw dose to avoid toxicity due to the impaired immune system. As evidenced by Figure 2B, there was no overall weight loss as a result of the injections of the compounds. Additionally, all mice exhibited normal behavior suggesting a lack of distress.

A375 CDX was established by injection of A375 cells into the right flank of the animals on day 9 after the arrival of mice. The tumors became palpable within two weeks after the injections. However, several mice did not develop tumors and were discarded from the study leaving 11 mice in each group. The tumors reached 100 mm^3^ on day 29 after which we commenced the treatment. The efficacy of 2155-14 and 2155-18 became evident one week after the start of the treatment (Figure 3A). We continued the treatment for two weeks until the tumors of the vehicle control group reached the maximum allowed volume, after which we discontinued the study. The tumor volumes of the treated groups were statistically significantly smaller than the vehicle control group throughout the entire study. On the final day of the study, the tumor volume ratio between the treated and vehicle control groups reached 10-fold. To confirm this observation, we excised and weighed the tumors. As can be seen from Figure 3B, the weights of the tumors from the treated groups were statistically significantly smaller than the vehicle control group. Interestingly, three tumors from 2155-18-treated mice were not found upon necropsy, whereas all tumors from the 2155-14-treated group were accounted for.

The excised tumors were stained for cleaved caspase 3 (CC3), an apoptosis marker, Ki67, a proliferation marker, and hnRNPH1/H2. Tumors treated with vemurafenib, 2155-14, and 2155-18 were positive for CC3 staining (Figure 4) consistent with their reported mechanism of cell death [8,19]. Tumors treated with 2155-14 and 2155-18 revealed decreased Ki67 staining similar to the tumors treated with vemurafenib, suggesting that cell proliferation is inhibited by the lead compounds (Figure 5).

Immunofluorescent staining of tumors demonstrated a significant decrease of hnRNPH1/H2 in vemurafenib-treated samples and the total loss of staining in the tumors treated with both 2155-14 and 2155-18 (Figure 6). This agrees with the hypothesis that downregulation of hnRNPH1/H2 can lead to the selective death of melanoma cells as we previously posited [8].

## 4. Discussion

The results presented in this study demonstrate the efficacy of two novel anti-melanoma leads, 2155-14 and 2155-18, in the A375 cell line-derived xenograft model. More specifically, the leads stopped tumor growth by blocking cell proliferation and caused apoptosis. To the best of our knowledge, this is the first example of compounds belonging to the pyrrolidine diketopiperazine chemotype demonstrating an anti-cancer activity.

Our group has previously shown that 2155-14 binds to spliceosomal proteins hnRNP H1 and H2 [8] causing ER stress and autophagy ultimately resulting in apoptotic cell death. This is, to our knowledge, the first and so far, the only report of small molecules working via binding the above-mentioned molecular targets. hnRNP H1 and H2 are RNA-binding proteins involved in mRNA splicing, export, and stability in normal biology [20]. Nothing is known about the role of hnRNPH1 and H2 in melanoma development and homeostasis. In other cancers, hnRNPH1 was shown to be upregulated in chronic myeloid leukemia (CML) patients and cell lines which correlated with disease progression [21]. In the same study, in vivo and in vitro experiments showed that the knockdown of hnRNP H1 inhibited cell proliferation and promoted cell apoptosis in CML cells. hnRNP H1 was also demonstrated to promote colorectal cancer progression via the stabilization of mRNA of Sphingosine-1-Phosphate Lyase 1 in vitro [22].

Mechanistically, the study by Uren et al. [23] identified a set of 1086 high-confidence target transcripts of hnRNP H1. Analysis of the target transcripts indicated that hnRNP H1’s involvement in splicing is 2-fold: it directly affects a substantial number of splicing events, but also regulates the expression of major components of the splicing machinery and other RNA-binding proteins (RBPs) with known roles in splicing regulation. The identified mRNA targets displayed function enrichment in MAPK signaling and ubiquitin-mediated proteolysis, which might be the main routes by which H1 promotes tumorigenesis. However, this study was conducted in HeLa cells, and it is not clear whether hnRNP H1’s role in melanoma cells is similar to its role in HeLa cervical carcinoma cells.

We recently demonstrated that siRNA-mediated knockdown of target proteins of 2155-14 and 2155-18, hnRNP H1 and H2, also leads to the increase in interferon signaling in the melanoma cell line WM266-4 [24]. We also demonstrated that hnRNP H1 and H2 are downregulated in WM266-4 cells after treatment with 2155-14 [8]. This suggests that the 2155-14 and 2155-18 mechanisms of action against A375 CDX can be based on the downregulation of hnRNP H1 and H2 leading to the increased immune response and anti-tumor immunity. Interestingly, a recent clinical informatics study by Kim et al. [25] using the data set from TCGA found that the increased expression of immune-related pathways (PD-1, interferon-α/β, and interferon-γ) correlates with increased overall survival of melanoma patients. This is important because PD-1, interferon-α/β, and interferon-γ pathways are associated with higher immune cell infiltration rates into tumors [26,27,28] leading to better response to immunotherapy.

We utilized nude athymic mice (nu/nu) for this study that are T-cell deficient [29]; however, they have partially functional B cells, functional NK cells, and macrophages, therefore, a contribution of the immune system to the anti-tumor activity of 2155-14 and 2155-18 cannot be ruled out at this stage. Possible activation of immune response by 2155-14 and 2155-18 against melanoma will have to be further studied using mice with intact immune systems.

Additionally, other groups reported that treatment of triple-negative breast cancer cells with spliceosomal inhibitors sudemycin D6 (SD6) and H3B-8800 resulted in increased immune signaling including interferon α and β pathways [30], which corroborates the hypothesis that spliceosomal modulation can lead to the increased immunogenicity of cancer cells.

## 5. Conclusions

The results of this study in combination with a previous report [10] of a lack of in vivo toxicity suggest that 2155-14 and 2155-18 can be used as drug development leads for melanoma therapy and in vivo and in vitro probes for melanoma research. Additionally, targeting hnRNP H1 and H2 represents a feasible approach to melanoma therapy both as monotherapy and as an adjuvant to immunotherapy.

## Data Availability

Not applicable.

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
