# Peer review of "Novel Anti-Melanoma Compounds Are Efficacious in A375 Cell Line Xenograft Melanoma Model in Nude Mice"

_biomolecules, 2023, doi:10.3390/biom13091276_

Round 1
Reviewer 1 Report
In this manuscript, authors proposed two new compounds (2155-14 and 2155-18) effective against xenograft models of melanoma. Authors showed that these two compounds reduce the A375 xenograft tumor growth in nude mice by activating caspase-3. Authors further showed that ki-67 and proposed target of 2155-14 and 2155-18 hnRNPH RBP was downregulated in these tumors. However, there are few shortcomings which needs to be addressed.
1. There is no mention of Figure 5 in Result section.
2. Why authors chose to target hnRNPH1 and H2 RBPs in melanoma? As authors themselves mentioned in the discussion section that the significance of these proteins have not been established in melanoma. Authors may show TCGA of analysis of any melanoma dataset to highlight the significance of these RBPs in melanoma which makes this manuscript complete.
3. In figure 5, the images shown for DAPI and overlay are at different magnifications. Please correct. Also, H&E images are not required as authors already provided the H&E images in Figures 3 and 4.
4. The selection of shown regions in Figure 5 may be improved especially for vemurafenib and 2155-14 groups.
5. How the size of 1X images for vehicle control and treated groups shown in Figures 3 and 4 are same. As shown in figure 2D, the treated tumors were very small in size compared to vehicle treated.
6. Please clarify how authors established xenograft tumors. Did authors used any matrix like matrigel? What was the site of injection?
7. What was the route of compound administration? Please mention in methods section.
8. Page 3, line 123- The HPLC flow rate has been mentioned at 60 mL/min. Please check.
9. Page 4, lines 158, 159. 2155-14 and 2155-18 repeated twice.
10. Page 5, line 208. Replace ki67 with Cleaved Caspase 3. Also change the title of section 2.5 to Cleaved-Caspase 3 staining.
Author Response
We thank Reviewer 1 for the constructive critique.
Our responses are as follows:
- There is no mention of Figure 5 in Result section.
Response. We added text referring to the Fig. 5 (lines 312-316 of the revised manuscript).
- Why authors chose to target hnRNPH1 and H2 RBPs in melanoma? As authors themselves mentioned in the discussion section that the significance of these proteins have not been established in melanoma. Authors may show TCGA of analysis of any melanoma dataset to highlight the significance of these RBPs in melanoma which makes this manuscript complete.
Response. We added a figure describing correlation of high expression of hnRNP H1 and H2 with low disease-specific survival probability (Fig. 1 of revised manuscript).
- In figure 5, the images shown for DAPI and overlay are at different magnifications. Please correct. Also, H&E images are not required as authors already provided the H&E images in Figures 3 and 4.
Response. We checked and confirmed that DAPI and overlay images are at the same magnification. H&E images in Fig. 5 and 3-4 were obtained using different embedding and cutting procedures (cryo for Fig.5 and 4% paraformaldehyde/paraffin for Figs. 3-4), therefore, H&E images are necessary for Fig.5 as well as for Figs. 3-4.
- The selection of shown regions in Figure 5 may be improved especially for vemurafenib and 2155-14 groups.
Response. Can the reviewer be more specific as to what they would like to see in the images?
- How the size of 1X images for vehicle control and treated groups shown in Figures 3 and 4 are same. As shown in figure 2D, the treated tumors were very small in size compared to vehicle treated.
Response. The tumors of vehicle control and treated groups shown in Figures 3 and 4 in some cases were cut along the long side of the tumors, which are oblong in shape (please see Fig. 2) which in these cases approximates the size/length of the control tumors, which creates the impression that they are of the same size. In other cases, the tumor samples from treated animals were cut along the short side and it can be seen from Fig 3 and 4 2155-14 that they are significantly smaller than the control samples.
- Please clarify how authors established xenograft tumors. Did authors used any matrix like matrigel? What was the site of injection?
Response. No matrix was used. A375 CDX was established by injection of A375 cells into the right flank of the animals on day 9 after the arrival of mice. This text was added on line 241 of the revised manuscript in the Results section. This was also mentioned in the Methods on line 153.
- What was the route of compound administration? Please mention in methods section.
Response. All compounds were injected subcutaneously (sc). The text was added to Methods on line 166-167.
- Page 3, line 123- The HPLC flow rate has been mentioned at 60 mL/min. Please check.
Response. The rate of 60 mL/min is correct. The preparative HPLC with preparative column was used which allows rates up to 100 mL/min.
- Page 4, lines 158, 159. 2155-14 and 2155-18 repeated twice.
Response. The duplicate text was deleted.
- Page 5, line 208. Replace ki67 with Cleaved Caspase 3. Also change the title of section 2.5 to Cleaved-Caspase 3 staining.
Response. ki67 was replaced with Cleaved Caspase 3 on line 209 of the revised manuscript and the title of section 2.5 was changed to Cleaved-Caspase 3 staining.
Reviewer 2 Report
The article under review is about anti-tumor activity of new pyrrolidine diketopiperazine derivatives in a xenograft model of melanoma. The article shows new targets and offers new strategies for melanoma therapy.
The article is quite short, descriptive, non-mechanistic but it is well-written, easy to follow, the main conclusions are supported by the results, and the methods are described properly.
The article is recommended for publication after minor revision.
My points are:
1. You have shown that in some melanoma cell lines 2155-14 cause ER stress, autophagy and mitochondrial apoptosis. Is there any signs of autophagy in case of A375 CDX?
2. Put your figures within the text of the manuscript, not in separate paragraph of the results section.
3. Describe your figure 5.
4. Introduction: give more information about your compounds - how did you designed that molecules, how you showed their targets, what are their EC50 in cell lines. The characterization of the molecules by NMR and HPLC can be showed as supplementary figure.
5. The discussion is short and gives little information about anti-tumor mechanisms of 2155-14/18. The possible mechanisms of anti-tumor activity of 2155-14/18 should be discussed (refs 8, 10.33594/000000164, , and so on), describe the possible action of compounds on the immune system (10.1158/1538-7445.AM2023-LB239), note that nu/nu mice still possess NK cells.
6. Justify the choice of 2155-14/18 doze.
7. Minor errors: italicize in vivo (line 25), "lack of clinical and organ toxicity" (line 235) - the toxicity in mice is not actually "clinical", also the "lack of toxicity" sounds strange, please rephrase, "melanoma actives" (line 23)- consider changing to "anti-melanoma compounds", title "anti-melanoma leads" (line 2) - consider rephrasing to "anti-melanoma compounds".
Generally, the article is interesting and should be accepted after minor changes.
Author Response
We thank Reviewer 2 for the constructive critique.
Our responses are as follows:
- You have shown that in some melanoma cell lines 2155-14 cause ER stress, autophagy and mitochondrial apoptosis. Is there any signs of autophagy in case of A375 CDX?
Response. We have not performed an autophagy analysis on the excised tumors.
- Put your figures within the text of the manuscript, not in separate paragraph of the results section.
Response. We put all figures within the text of the manuscript.
- Describe your figure 5.
Response. We added text referring to the Fig. 5 (lines 312-316 of the revised manuscript).
- 4. Introduction: give more information about your compounds - how did you designed that molecules, how you showed their targets, what are their EC50 in cell lines. The characterization of the molecules by NMR and HPLC can be showed as supplementary figure.
Response. As mentioned in the original manuscript, the characterization of the molecules by NMR and HPLC has been previously published by us and, therefore, is not necessary for this manuscript. The text describing compound discovery, target identification, and cell line profiling data has been added to the introduction of the revised manuscript using track changes.
- The discussion is short and gives little information about anti-tumor mechanisms of 2155-14/18. The possible mechanisms of anti-tumor activity of 2155-14/18 should be discussed (refs 8, 10.33594/000000164, , and so on), describe the possible action of compounds on the immune system (10.1158/1538-7445.AM2023-LB239), note that nu/nu mice still possess NK cells.
Response. Little information about MoA of 2155-14/18 is provided for the simple reason that we don’t know much about it. We incorporated the discussion points suggested by the reviewer to the revised manuscript.
- Justify the choice of 2155-14/18 doze.
Response. We added justification in the Results section in the “In vivo efficacy study in A375 CDX” sub-section of the revised manuscript.
- Minor errors: italicize in vivo (line 25), "lack of clinical and organ toxicity" (line 235) - the toxicity in mice is not actually "clinical", also the "lack of toxicity" sounds strange, please rephrase, "melanoma actives" (line 23)- consider changing to "anti-melanoma compounds", title "anti-melanoma leads" (line 2) - consider rephrasing to "anti-melanoma compounds".
Response. We implemented these edits – please see them in track changes in the revised manuscript.
Round 2
Reviewer 1 Report
The addition of Figure 1 improved the paper. Authors addressed all other concerns satisfactorily.
Regarding my comment 3 in previous round of review, the magnifications for the DAPI and Overlay images for 2155-14 group in Figure 6 (previously figure 5) are indeed different. The DAPI image is of higher magnification than overlay.